# Atoms to Events: Categorical Evidence Composition for Video Anomaly Detection

## Abstract

Video anomaly detection (VAD) seeks to identify events that deviate from learned normality. Current Vision–Language Models (VLMs) face significant challenges: anomalies are rare, labels are weak, and visual appearance varies drastically. Mainstream VLMs directly map visual features to events, they overfit to intermediate incidental cues which are present during training and generalize poorly. To address this issue, we propose a categorical view of anomaly understanding. Firstly, an Unsupervised Anomalous Period Detector (UAPD) is proposed to identify abnormal periods. Next, a Category-based Atom Miner (CAM) is proposed to map visual features to learned atoms in video segments, and learn the roles of atoms. In inference, CAM provides role-aware indications to VLM which maps meaningful atoms and visual features to event predictions. This framework harnesses meaningful evidence and preserves the generalization capacity of VLMs. Extensive experiments and ablations show consistent gains over strong vision-only and fine-tuned VLM baselines.

## 1 Introduction

Video anomaly detection (VAD) seeks to localize the time spans in long videos where scenes deviate from regular patterns—e.g., violence, accidents, explosions, or other unexpected events. The problem is challenging because anomalies are rare in contrast to the massive scale of surveillance streams, visual conditions vary substantially across cameras and over time due to viewpoint, lighting and other factors, and labels are typically video-level or even abscent. Existing approaches have explored vision-only strategies that rely purely on visual features: prediction-based models that forecast future sequences and flag deviations, reconstruction-based models that assume anomalies reconstruct poorly, representations combining multiple feature types or Multiple Instance Learning (MIL)-based approaches Li et al. (2022); Georgescu et al. (2021); Park et al. (2020); Noghre et al. (2024); Yang et al. (2023); Liu et al. (2021); Huang et al. (2025); Georgescu et al. (2021); Cho et al. (2023); Guo et al. (2023); Liu et al. (2022). Although these approaches are effective, some of them cannot perform well under domain shifts, and may mistake irrelevant variations for true anomalies.

To improve interpretability and zero-shot generalization, recent works leverage Vision–Language Models (VLMs). They use captions, prompts, or pseudo-labels to score abnormal semantics within sliding windows Cao et al. (2024); Chen et al. (2024); Yang et al. (2024a); Micorek et al. (2024); Zhu & Pang (2024); Li et al. (2024); Yang et al. (2024b); Tang et al. (2025). Training-free approaches Zanella et al. (2024) directly caption frames with VLM and summarize how captions change over time. Guidance-driven methods focus VLMs on anomaly evidences by leveraging verbalized questions Ye et al. (2025) or sampling the most suspicious snippets Zhang et al. (2025). These methods are interpretable. However, VLM-based detectors often encode an event as one entangled concept, letting nuisance factors, such as viewpoints and backgrounds, swamp model representations. As a result, the visual features of events are not well distinguished.

To stably attend VLMs to anomaly-relevant semantics without being distracted by variations, challenges lie in the set-theoretic fashion of VLMs. They learn a direct mapping from visual features to event labels by accumulating "relevant" cues, a video is treated as a set of features whose memberships vote for the label. Such membership often admits incidental cues and spurious contexts. We find that meaningful features can be divided into directly relational, inherently relational and counter evidences. In this regard, we propose a categorical, role-aware view: we treat atoms as objects with

roles in events — Direct(support), Indirect (synergy), and Counter(inhibition). The "visual features to events" mapping in set-theoretic VLMs changes to "visual features to atoms to events". The cues without valid roles have no effect. Robust reasoning and auditable explanations are yielded.

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
