Hang Zhou, Junqing Yu, and Wei Yang. Dual memory units with uncertainty regulation for weakly supervised video anomaly detection. *arXiv preprint arXiv:2302.05160*, 2023a.

Qihang Zhou, Guansong Pang, Yu Tian, Shibo He, and Jiming Chen. Anomalyclip: Object-agnostic prompt learning for zero-shot anomaly detection. *arXiv preprint arXiv:2310.18961*, 2023b.

Yixuan Zhou, Yi Qu, Xing Xu, Fumin Shen, Jingkuan Song, and Heng Tao Shen. Batchnorm-based weakly supervised video anomaly detection. *IEEE Transactions on Circuits and Systems for Video Technology*, 2024.

Jiawen Zhu and Guansong Pang. Toward generalist anomaly detection via in-context residual learning with few-shot sample prompts. In *Proceedings of the IEEE/CVF Conference on Computer Vision and Pattern Recognition*, pp. 17826–17836, 2024.

# A    DETAILED STRUCTURE OF CGSGM

To learn the interplay among semantic components, CGSGM learns from normal training data the way that $\mathbf{G}_i^1(t), \mathbf{G}_i^2(t), \mathbf{G}_i^3(t), \mathbf{G}_i^4(t)$ combine into global semantic $\mathbf{G}_i^5(t)$. Specifically, we initialize CGSGM's input token sequence as $[\mathbf{S}_{i,1}(t), \mathbf{0}_{768}]$ where $\mathbf{S}_{i,1}(t)$ is the special token "[BOS]" signaling the start of sequence generation process Radford et al. (2018) Devlin et al. (2018). $\mathbf{0}_{768}$ is a 768-dimensional zero placeholder. $[\mathbf{S}_{i,1}(t), \mathbf{0}_{768}]$ is embedded into $[\mathbf{E}_{i,1}^T(t) \in \mathbb{R}^{H_d}, \mathbf{E}_{i,2}^T(t) \in \mathbb{R}^{H_d}]$ which is concatenated with $[\mathbf{G}_i^1(t), \mathbf{G}_i^2(t), \mathbf{G}_i^3(t), \mathbf{G}_i^4(t)]$, producing the input $\mathbf{X} = [\mathbf{G}_i^1(t), ..., \mathbf{G}_i^4(t), \mathbf{E}_{i,1}^T(t), \mathbf{E}_{i,2}^T(t)]$ to CGSGM's masked self-attention layer.

As is shown in Fig. 4, the mask in self-attention layer enables the tokens of local components to attend to each other, and facilitates the global semantic tokens to attend to all local tokens. Provided with input sequence $\mathbf{X}$, query, key and values are $\mathbf{Q} = \mathbf{X}\mathbf{W}^\mathbf{Q}$, $\mathbf{K} = \mathbf{X}\mathbf{W}^\mathbf{K}$ and $\mathbf{V} = \mathbf{X}\mathbf{W}^\mathbf{V}$ with $\mathbf{W}^\mathbf{Q}$, $\mathbf{W}^\mathbf{K}$ and $\mathbf{W}^\mathbf{V}$ being learnable weights, self-attention is implemented by

$$\mathbf{Z}_i^T(t) = Softmax(\frac{\mathbf{Q}\mathbf{K}^\mathbf{T}}{\sqrt{H_d/h}} + \mathbf{M})\mathbf{V} \tag{11}$$

where $\mathbf{M}$ denotes mask. $h = 12$ denotes the number of heads. The output of masked self-attention layer is denoted as $\mathbf{Z}_i^T(t) = [\mathbf{Z}_{i,1}^T(t), ..., \mathbf{Z}_{i,J-1+S_l}^T(t)]^T \in \mathbb{R}^{(J-1+S_l) \times H_d}$, $H_d = 768$. $J - 1 = 4$ and $S_l = 2$ denote the number of local components and the initialized sequence length, respectively. Only the last token $\mathbf{Z}_{i,J-1+S_l}^T(t)$ is fed into feed-forward layer because the last token is informative about the complete sequence of local components. The feed-forward layer has $H_d$ input channels and $H_d$ output channels. The output $\hat{\mathbf{S}}_{i,2}(t)$ denotes the embedding of generated global semantic $\hat{\mathbf{G}}_i^5(t) = \hat{\mathbf{S}}_{i,2}(t)$.

CGSGM is trained with objective $-\log \left\langle \hat{\mathbf{G}}_i^5(t), \mathbf{G}_i^5(t) \right\rangle$ where $\left\langle \hat{\mathbf{G}}_i^5(t), \mathbf{G}_i^5(t) \right\rangle$ is cosine similarity. The learning rate schedule is Linear Warmup With Cosine Annealing. The warmup learning rate is $10^{-6}$ which increases to initial learning rate $10^{-4}$ and then decreases to minimum learning rate $10^{-5}$ in a cosine annealing learning rate schedule. The warmup stage lasts for 5000 steps. The batch size for training is 120. CGSGM is trained only on normal videos, the ground truth texts for training CGSGM are extracted from training data with VLM Wang et al. (2024).

# B    EXAMPLE ATOMS

In this section the atoms on XD-Violence with direct and indirect roles are provided as examples. An event is considered present if at least one direct atom is observed, or if at least two indirect atoms are observed. A counter case occurs when no direct atoms are observed and at most one indirect atom is present. The atoms are translated from explicitly violent language to common language using Wang et al. (2024).

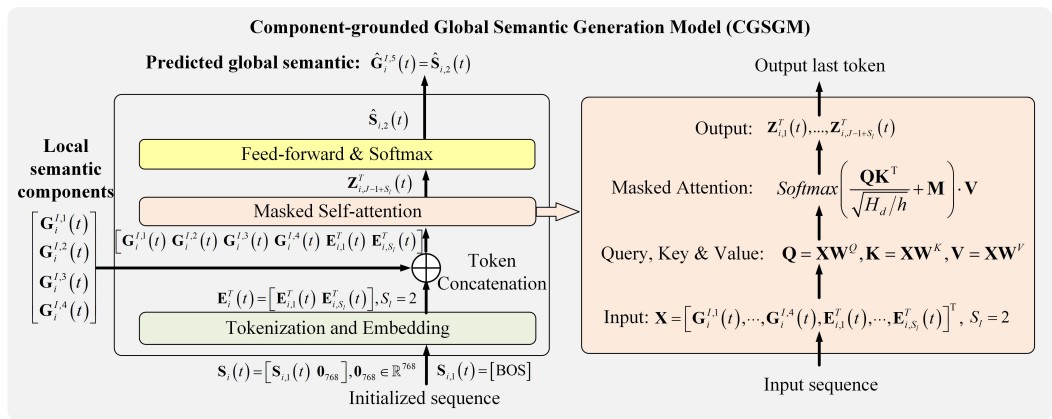

Figure 4: Detailed structure of CGSGM.

Table 3: Atoms of different anomalies

| Anomaly | Cue | Atoms | Descriptions in Common Language |
|---|---|---|---|
| Shooting | Direct | Muzzle flash | Sudden burst of light coming from the end of a moving object. |
| Shooting | Direct | Ejected shell | Small object rapidly propelled into the air. |
| Shooting | Direct | Smoke at muzzle | Smoke emerging from the end of a moving object. |
| Shooting | Direct | Barrel recoil | A mechanical component moving backward quickly. |
| Shooting | Direct | Arms or shoulders in immediate recoil, muzzle visible | Person's shoulders and body recoiling while holding an object. |
| Shooting | Indirect | Bullet impacts dust or debris | Dust and debris appear on a surface. |
| Shooting | Indirect | Small star-like flash | Small flash in the background. |
| Shooting | Indirect | Thin luminous streaks | Thin glowing streaks moving in a straight line. |
| Shooting | Indirect | Spark on surface | Spark on a surface. |
| Shooting | Indirect | Person's sudden pain reaction | People react to a sudden force on their body. |
| Shooting | Indirect | People aiming or holding guns | Multiple individuals holding objects and aiming at something. |
| Shooting | Indirect | Holes on surfaces | Holes on walls or panels. |
| Accident | Direct | Vehicles in hard contact | Two vehicles in contact at an impact point with deformation. |
| Accident | Direct | Debris flying from vehicle | Parts, glass, dust flying from the impact point. |
| Accident | Direct | Airbag inflating | Airbag inflating in this frame. |
| Accident | Direct | Skid marks terminate at impact point | Skid marks ending at the impact point of vehicle. |
| Accident | Direct | Vehicle strikes an object with deformation | A vehicle actively strikes an object with deformation at the strike point. |
| Accident | Indirect | Structural damage | Broken hood and door, loose parts and shattered glass on the road nearby. |
| Accident | Indirect | Deployed airbag open | Airbag deployed. |
| Accident | Indirect | Smoke or fluid leak from vehicle with crash | Smoke, steam or fluid leak from a crashed vehicle. |
| Accident | Indirect | Vehicle in abnormal orientation or position | Vehicles stopped in abnormal orientation or location. |
| Accident | Indirect | Skid marks on road | Skid marks near damaged vehicles. |
| Accident | Indirect | Impact mark with debris | Fresh mark on guardrail or wall with debris around. |
| Accident | Indirect | Emergency responder, damage visible | Emergency responders and damage. |
| Explosion | Direct | Intense bright flash | Sudden bright burst or intense spherical flash. |
| Explosion | Direct | Radial ejecta | Debris and dust visibly flying outward from a central point. |
| Explosion | Direct | Blast ring | Expanding circular dust front from a center. |
| Explosion | Direct | Ignitions | Multiple ignitions starting near the same central origin. |
| Explosion | Direct | Windows or panels shatter outward | Windows or panels actively shattering outward, shards moving away from a source. |
| Explosion | Indirect | Radial damage pattern | Blown-out window and doors or facade peeled outward. |
| Explosion | Indirect | Crater or scorched debris | Scorched epicenter with surrounding debris field. |
| Explosion | Indirect | People or objects thrown away | Objects or people thrown or falling away from a central point. |
| Explosion | Indirect | Smoke cloud and blast | Large smoke cloud consistent with a recent blast. |
| Explosion | Indirect | Panels torn outward and dense fresh soot plume | Vehicle or building panels torn outward with smoke. |
| Explosion | Indirect | Multiple fires | Multiple fires near the same origin. |
| Fighting | Direct | Strike with arm making contact | Hand or arm making contact with a person, with impact. |
| Fighting | Direct | Kick a person | Leg making contact with a person. |
| Fighting | Direct | Grappling and throwing | Hands gripping clothing, one person lifting another. |

Table 3 (continued)

| Anomaly | Cue | High-frequency Atoms | Detailed Description |
|---|---|---|---|
| Fighting | Direct | Forceful shove | Forceful shove that visibly displaces person's body. |
| Fighting | Direct | Person striking another with a hand-held object | A handheld object is used to strike a person. |
| Fighting | Indirect | Raised fists or wind-up posture at close range facing an opponent | Raised fists or wind-up posture at close range facing another person. |
| Fighting | Indirect | Person falling, stumbling back | A person falling, stumbling back while others advance toward them. |
| Fighting | Indirect | Clothing or hair being pulled, pain reaction during scuffle | clothing or hair being pulled, with pain reaction during a scuffle. |
| Fighting | Indirect | Security separating people, stepping in with urgent gestures | Security stepping in with urgent gestures to separate people. |
| Fighting | Indirect | Objects thrown toward a person | An object is thrown toward a person from a scuffle. |
| Fighting | Indirect | Multiple people converging aggressively | Multiple people converging on one person in a way suggesting conflict. |
| Fighting | Indirect | Holding weapon | A person is holding a weapon. |
| Riot | Direct | People clashing with police | People pushing or in contact with security personnel. |
| Riot | Direct | Rocks, bottles, fireworks thrown in the air | Rocks, bottles, fireworks thrown toward people or vehicles or buildings. |
| Riot | Direct | Property destruction | Breaking windows or doors. |
| Riot | Direct | People taking goods from a damaged store | People removing goods from a damaged storefront. |
| Riot | Direct | People igniting objects or vehicles | A person igniting objects or vehicles, flames beginning. |
| Riot | Indirect | Broken windows with glass scattered | Broken windows or doors with glass scattered. |
| Riot | Indirect | Vehicles overturned, damaged | Vehicles damaged, overturned or on fire. |
| Riot | Indirect | Large fire or thick smoke cloud in a street scene with a crowd | Large fire or smoke cloud in a street scene where a crowd is visible. |
| Riot | Indirect | Barricades across road | Barricades spanning a route. |
| Riot | Indirect | People carrying improvised weapons and shields | People carrying sticks, bricks or shields. |
| Riot | Indirect | Aggressive people rushing | People rushing |
| Riot | Indirect | Widespread debris and signage damage, unrest visible | Debris and damage with unrest. |
| Abuse | Direct | Strike in contact to a person | Hit a person. |
| Abuse | Direct | Hands and arm around neck | Hands or arms around neck. |
| Abuse | Direct | Prevent people's movement | Person restrained, pressed to wall, preventing movement. |
| Abuse | Direct | Drag by limb or clothing | Hair pulled in contact, or person dragged. |
| Abuse | Direct | Hit people with an object | Object directed at a person and used to hit. |
| Abuse | Indirect | Defensive posture | Arms shielding face, recoiling from another person. |
| Abuse | Indirect | One person approaches the other cornered person | One person looming, the other cornered. |
| Abuse | Indirect | Injury on body | Injury or impact puff on body. |
| Abuse | Indirect | Person being pulled | Person pulled. |
| Abuse | Indirect | Object thrown toward a person | Object thrown toward a person. |
| Abuse | Indirect | Pleading gestures | Hands up, shielding self. |
| Abuse | Indirect | Multiple people advancing aggressively toward one individual | Multiple people advancing toward a single person. |

## C STRUCTURE OF VISUAL ENCODER

Section 3.2 presents a visual encoder $E_V$ which encodes frames $\{t, ..., t+l-1\}$ in Clip $t$ to a feature vector $\mathbf{v}_t \in \mathbb{R}^{d_v}, d_v = 768$. Specifically, we firstly leverage "ViT-L/14" model Radford et al. (2021) in encoding all $l$ frames in a clip to feature vectors $\{f_t, ..., f_{t+l-1}\}, f_\tau \in \mathbb{R}^{d_v}, \tau = t, ..., t + l - 1$. With the same structure as CGSGM, $E_V$ takes in input sequence $f_t, ..., f_{t+l-1}$ and predicts $\hat{\mathbf{v}}_t$. The objective for training is $-\log \langle \hat{\mathbf{v}}_t, E_T(e(t)) \rangle$ based on the cosine similarity between prediction $\hat{\mathbf{v}}_t$ and the embedding of event label $e(t)$ at $t$. The text embedding is provided by text encoder $E_T$. In implementations, we firstly train CGSGM before detecting abnormal periods. Then leverage the detections results for training $E_V$, using the same hyperparameters as CGSGM. Finally, $E_V$ is frozen before training CAM.

## D FINE-TUNING VLM

We start from Qwen2.5-VL-7B-Instruct and apply LoRA *adapters* to the language-model blocks while keeping the vision tower frozen. We fine-tune using short-video clips: each sample provides $l = 8$ uniformly sampled frames at $224 \times 224$ and a chat-style instruction "Please classify the event

Table 4: Performance on Ubnormal (AUC, %) and NWPU (AUC, %).

| Ubnormal | | NWPU | |
|---|---|---|---|
| **Method** | **AUC** | **Method** | **AUC** |
| Hirschorn et al.Hirschorn & Avidan (2023) | 79.2 | Cao et al. Cao et al. (2023) | 68.2 |
| Micorek et al. Micorek et al. (2024) | 72.8 | Zhang et al. Zhang et al. (2024b) | 67.3 |
| Yang et al. Yang et al. (2024a) | 71.9 | | |
| **Ours** | **79.6** | **Ours** | **68.8** |

in this clip." to classify the clip into one event type. The tokenizer maximum length is 1024 text tokens. Training uses a per-GPU batch size of 2 with gradient accumulation to reach an effective batch size of 64, for 5 epochs. We employ AdamW with $(\beta_1, \beta_2) = (0.9, 0.999)$ and weight decay 0.05. The learning rate for LoRA parameters is $1 \times 10^{-4}$ with a 5% warm-up and cosine decay. Precision is `bfloat16` with gradient checkpointing enabled. LoRA configuration includes: rank $r = 16$, $\alpha = 32$, dropout is 0.05, there is no bias. The objective is token-level cross-entropy. Frame shuffling is disabled to preserve temporal order.

# E  EXPERIMENTS ON ADDITIONAL DATASETS

To evaluate the capability of the approach on generalization, we evaluate it on NWPU Cao et al. (2023) and Ubnormal Acsintoae et al. (2021). NWPU includes 305 training videos and 242 testing videos. In the training data, there are only normal events, while the test data contains both normal events and anomalous events. In the test set, there are 28 classes of abnormal events, frame-level labels indicating whether each frame is normal or abnormal are provided. Ubnormal is divided into a training set with 268 videos, a validation set with 64 videos, and a test set with 211 videos. All three sets include normal and abnormal events. All videos are with frame-level labels. Among the 22 types of anomalies, 6 are present in training set, 4 for validation and 12 in test set. The performance is shown in Table 4.

On Ubnormal, UAPD and CAM-guided VLM are leveraged. Although the anomaly types in training, validating and testing data are different, they share commonalities in terms of atoms. Besides, we only care about the discrimination between normal and abnormal events, the confusion in anomaly types does not matter. So the performance of the proposed approach is good. On NWPU, we leverage "Setting 1" in Table 2, keeping only UAPD and a classifier because the training set only includes the videos without anomalies. The proposed UAPD achieves state-of-the-art performance.

# F  SUBJECTIVE RESULTS

Fig.5 shows subjective results illustrating the determination of events based on the presence scores of atoms and VLM. One or more direct atoms support the presence of an event, two or more indirect atoms support the presence of an event. If no direct atom is present and no more than one indirect atom is present, then the event is not present.

# G  CROSS-DATASET EVALUATION

To validate generalization capability, we evaluate the performance of the proposed approach when being trained on XD-Violence and tested on UCF-Crime. Specifically, XD-Violence includes 6 types of anomalies while UCF-Crime includes 13 types. However, their semantic atoms share commonalities. In implementations, we train CAM on XD-Violence, and only change the prompt to VLM ("The presence of at least one direct or two indirect atoms supports an event. Determine whether any event in the set {Fighting, Shooting, Riot, Abuse, Car accident, Explosion} is present") to "The presence of at least one direct or two indirect atoms supports an event. Determine whether any event in the set {Abuse, Arrest, Arson, Assault, Burglary, Explosion, Fighting, Road Accident, Robbery, Shooting, Shoplifting, Stealing, Vandalism} is present" which corresponds to the anomaly types in

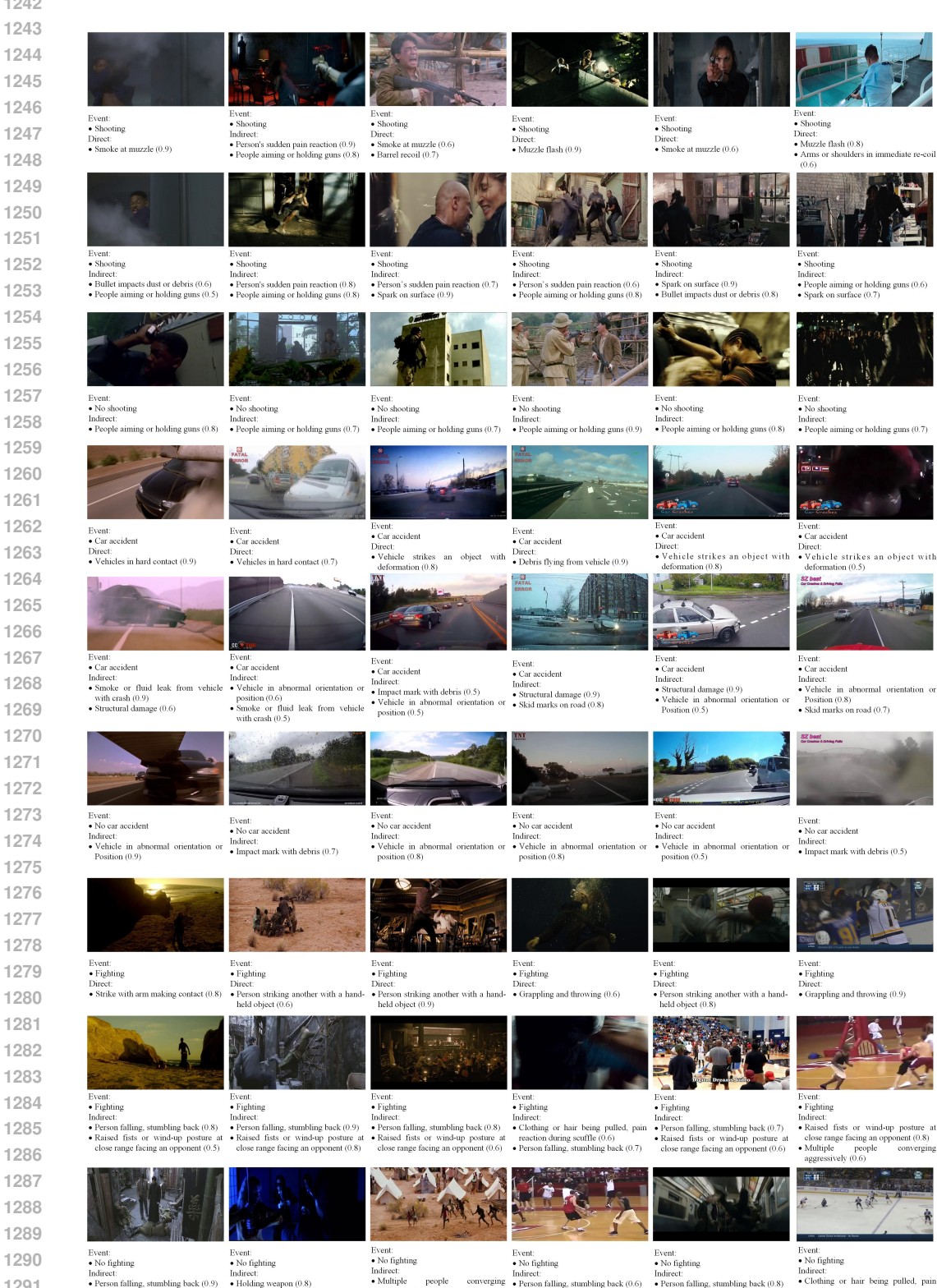

Figure 5: Subjective results of anomaly detection using atoms. The atoms are inferred from video segments.

Table 5: Cross-dataset evaluation. Train on XD-Violence and Evaluate on UCF-Crime (AUC, %).

| Settings | AP |
|---|---|
| Setting 1: Train on UCF-Crime, Evaluate on UCF-Crime | 91.42 |
| Setting 2: Train on XD-Violence, Evaluate on UCF-Crime | 88.36 |

UCF-Crime. In this way, VLM leverages CAM's indication about atoms in identifying the events in other types. As can be seen from Table 5, the performance does not change significantly.

## H ADDITIONAL ABLATION STUDIES

**Expressions of Atoms in Common Language** As is addressed in Section 3.2, we translate atoms from explicitly violent language to common language before determining the presence scores of atoms, because most of the models that compute image-text similarity have seldomly observed abnormal events during training. Specifically, we achieve this by prompting Qwen2.5-VL-3B Bai et al. (2023) with "Please describe this phrase in common language." By comparing Settings 1 and 2 in Table 6, we can see that the descriptions of atoms in common languages contribute.

**Number of Atoms as Prompt to VLM** As is addressed in Section 3.3, we generate prompts with the atoms that have $top-C$ highest presence scores as prompt to guide VLM. By comparing Settings 1, 3 and 4 in Table 6, we can see that $C = 10$ is an appropriate choice.

**Number of Clips per Video for Atom Mining** As is addressed in Section 3.2, for mining atoms, we search from the clips with $top-K$ highest anomaly scores in each training video. By comparing Settings 1, 5 and 6, we can see that $K = 20$ is an appropriate choice, a larger $K$ may introduce false alarms.

**Length of Video Clips** As is addressed in Section 3.2 and 4.1, we determine the presence of atoms in the unit of video clips each with length $l$. By comparing Settings 1, 7 and 8, we can see that $l = 8$ is an appropriate choice for extracting temporal features.

**Necessity of Mining Atoms from Data** As is addressed in Section 3.2, we mine atoms from abnormal clips. To validate the advantage of mining from data over mining from VLM's memory, we take the event "explosion" as an example, and replace the prompt "Please identify the semantic atoms that can infer explosion" with "Please identify the semantic atoms that can infer explosion, do not refer to input video." The results are shown in Setting 9 in Table 6. It can be seen that performance drops significantly.

**Necessity of Deduplication** As is addressed in Section 3.2, we prompt VLM with "Whether this atom has already been described by those in the sets" to only add novel atoms to sets. To validate the contribution of this part, we remove it and evaluate the performance in Setting 10 of Table 6. It can be seen that discriminative atoms improve performance by providing more information.

Table 6: Additional ablation studies on two benchmarks: UCF-Crime (AUC, %) and XD-Violence (AP, %).

| Settings | UCF-Crime | XD-Violence |
|---|---|---|
| Setting 1: UAPD + CAM-guided VLM, common language (Proposed), $C = 10, K = 20, l = 8$ | 91.42 | 91.09 |
| Setting 2: UAPD + CAM-guided VLM, original atoms without translation $C = 10, K = 20, l = 8$ | 88.65 | 86.52 |
| Setting 3: UAPD + CAM-guided VLM, common language, $C = 5, K = 20, l = 8$ | 90.25 | 90.02 |
| Setting 4: UAPD + CAM-guided VLM, common language, $C = 20, K = 20, l = 8$ | 91.38 | 90.89 |
| Setting 5: UAPD + CAM-guided VLM, common language, $C = 10, K = 10, l = 8$ | 91.41 | 90.97 |
| Setting 6: UAPD + CAM-guided VLM, common language, $C = 10, K = 40, l = 8$ | 83.01 | 81.27 |
| Setting 7: UAPD + CAM-guided VLM, common language, $C = 10, K = 20, l = 4$ | 90.89 | 90.75 |
| Setting 8: UAPD + CAM-guided VLM, common language, $C = 10, K = 20, l = 16$ | 91.43 | 91.11 |
| Setting 9: UAPD + CAM-guided VLM, common language, $C = 10, K = 20, l = 8$ | 73.56 | 71.28 |
| Setting 10: UAPD + CAM-guided VLM, common language, $C = 10, K = 20, l = 8$ | 82.21 | 81.09 |

# I  THE USE OF LARGE LANGUAGE MODELS (LLMs)

In this paper, we have not used LLMs in research ideation and writing. As a result, there's no issue concerning the usage of LLMs.