# OpenReview forum: "Atoms to Events: Categorical Evidence Composition for Video Anomaly Detection"
_ICLR.cc/2026/Conference — ICLR 2026 Conference Withdrawn Submission_

### Official Review · Reviewer_cKc1 · 2025-10-16

**Soundness:** 2
**Presentation:** 1
**Contribution:** 2
**Rating:** 2
**Confidence:** 5

**Summary:**

The authors propose an Unsupervised Anomalous Period Detector (UAPD) and a Category-based Atom Miner (CAM) to identify and learn the roles of direct, indirect, and counter evidence atoms. Guided by these atoms, a Vision–Language Model (VLM) performs interpretable and generalizable anomaly detection.

**Strengths:**

a. The overall detection performance is compelling and achieves SOTA results.

b. The framework combines unsupervised detection and language-guided reasoning, which is a promising direction for explainable video anomaly detection.

**Weaknesses:**

a. The presentation quality of the paper is poor. Many newly introduced concepts are not clearly defined or intuitively explained, making the methodology difficult to follow.

b. Figures are not clearly presented or discussed in the main text. Figure 1 is not referenced in the introduction, and its depiction of roles and relationships is unclear. Figure 2 also lacks clarity regarding the architecture and data flow — for example, it is not obvious what the three outputs of CAM represent, how CGSGM operates internally, or how the online and offline stages are connected.

**Questions:**

a. Figure 1 is presented early but not referenced or explained in the Introduction. How does it support the motivation of the framework?

b. The concept of an “Atom” should be clarified. What are its visual or semantic boundaries, and how are atoms identified in practice?

c. Please provide detailed and concrete examples of “Direct,” “Indirect,” and “Counter” atoms in real video contexts. How are these categories determined or learned?

d. How are anomaly scores computed? Are they produced only by UAPD, or further refined by CAM and VLM?

e. What is the final output of CAM — are these atom presence scores, event likelihoods, or another representation?

f. Please clarify the relationship between offline training and online inference. Is UAPD involved during inference, or only used for pre-segmentation during training?

g. In Figure 2, the input-output relationships among UAPD, CAM, and VLM should be explicitly shown. It is currently unclear how information flows across these modules

---

### Official Review · Reviewer_XACT · 2025-10-27

**Soundness:** 3
**Presentation:** 2
**Contribution:** 2
**Rating:** 4
**Confidence:** 3

**Summary:**

The paper introduces an “atoms-to-events” framework that decomposes video anomalies into interpretable evidence atoms before event prediction. It improves both accuracy and interpretability over existing vision–language and vision-only baselines.

**Strengths:**

- Originality: The “atoms-to-events” decomposition introduces a novel conceptual perspective for anomaly reasoning, providing a more interpretable and modular structure compared to conventional direct feature-to-event mappings. However, most components of the framework are adapted from existing architectures and methods, making the originality of the work lie primarily in its integration and system design rather than in fundamentally new algorithmic innovations.

- Quality: The framework achieves SOTA on both UCF-Crime and XD-Violence. The ablation studies in both main paper and appendix are thorough, and clearly show how atoms, segmentation, and the CAM–VLM integration each affect the results. Training details are extensive and reproducibility claims are strong.

- Clarity: The “atoms-as-evidence” perspective is well motivated and clearly justified through diagrams and textual expositions.

- Significance: This paper addresses an important challenge in video anomaly detection by proposing an interpretable “atoms to events” reasoning framework. The idea is promising and could influence future work on structured reasoning in vision–language models.

**Weaknesses:**

- The core novelty lies in framing, not architecture or algorithms.
- The “categorical” foundation is only loosely connected to actual category theory, making the theoretical justification seem superficial.
- Although qualitative results are shown, there is no quantitative validation of interpretability.
- Runtime and scalability analyses are absent.

**Questions:**

- Can authors provides with some quantitative validation of interpretability?
- How does the model handle overlapping or ambiguous events that may involve multiple atom types simultaneously?
- What is the computational cost of the full pipeline, and can it run efficiently on long or real-time video streams?

---

### Official Review · Reviewer_n466 · 2025-10-29

**Soundness:** 2
**Presentation:** 3
**Contribution:** 2
**Rating:** 2
**Confidence:** 5

**Summary:**

While this paper addresses a critical challenge in video anomaly detection (VAD)—mitigating VLM overfitting to spurious cues via a "visual features → atoms → events" framework—this paper suffers from insufficient methodological validation, limited experimental scope, and unaddressed practical limitations that undermine its scientific rigor and contribution to the field. Specifically,
1. Inadequate Validation of Core Methodological Components
2. The core experiments focus exclusively on UCF-Crime and XD-Violence, two well-studied but homogeneous datasets (primarily surveillance footage).
3. The paper’s qualitative results and ablation studies are insufficient to confirm the framework’s interpretability and core design choices.

In summary, this paper’s core idea of "atoms as intermediate evidence" is intriguing, but it is undermined by incomplete methodological validation, narrow experiments, and unaddressed practical limitations. The gaps in component ablation, SOTA benchmarking, and generalization testing mean the work does not meet ICLR’s standards for scientific rigor and impact. A reject decision is warranted.

**Strengths:**

This paper addresses a critical challenge in video anomaly detection (VAD)—mitigating VLM overfitting to spurious cues via a "visual features → atoms → events" framework

**Weaknesses:**

1. The proposed framework’s three pillars (UAPD, CAM, Atom-Guided VLM) lack rigorous validation of their individual and interactive efficacy, leaving critical questions about their necessity and design choices unresolved.
2. The experimental design is narrow, failing to demonstrate the framework’s robustness, generalizability, or superiority to state-of-the-art (SOTA) methods in diverse scenarios.
3. The core experiments focus exclusively on UCF-Crime and XD-Violence, two well-studied but homogeneous datasets (primarily surveillance footage). Appendix experiments on NWPU and Ubnormal are underdeveloped
4, Lack of Meaningful Qualitative Evaluation. Figures 3 and 5 show only 4 qualitative cases (shooting, accident, etc.) with no statistical validation. It does not report quantitative metrics for qualitative performance (e.g., intersection-over-union (IoU) between predicted and ground-truth anomalous intervals across the test set).
5. The paper’s atoms are tied to pre-defined anomaly types (shooting, explosion, etc.). It provides no experiments on unseen anomaly types (e.g., "crowd stampede," "fire outbreak")—a critical gap for generalizable VAD. Can the framework automatically mine new atoms for unseen anomalies, or does it require retraining?

**Questions:**

See the above comments

---

### Official Review · Reviewer_Rw8p · 2025-10-30

**Soundness:** 2
**Presentation:** 1
**Contribution:** 1
**Rating:** 2
**Confidence:** 4

**Summary:**

This work introduces to replace “visual features to event” mapping with “visual features to learnable atoms then to events”, a new framework for video anomaly detection.

The proposed method learns the atoms in video segments and the roles of atoms by incorporating the VLM for event predictions.

The evaluations are performed on two benchmark datasets and show improved performance.

**Strengths:**

+ The paper is technical sound.

+ In introduction section, this work is somehow well motivated with clearly listed contributions at the end.

+ The evaluations show improved performance on two datasets.

**Weaknesses:**

- Related work section listed many existing works; however, these works are not being discussed or analysed in this section. The challenges are outlined, but why existing methods are unable to address them, and how the proposed method addresses these are unclear. Also it would be better to outline how the proposed method differs from existing ones, and provide some insights. The current related work section needs significant revisions to reflect on these.

- Although the idea of using atoms for event predictions is interesting, (i) it would be better to have a notation detailing the maths symbols and operations used in the paper, eg, what are scalars, vectors and matrices etc. (ii) The overview of this work is not clearly and properly presented. This section is organised in an unnecessary complex way, and each part is quite standalone and does not very well connected to each other, this makes the assessments on novelty, core innovations harder.

- The authors introduced some new concepts; however, these concepts are not being clearly explained before using them. Also in the appendix, there is no enough information presented, making the understanding of the paper challenging. Some of the maths symbols are not being introduced before using them.

- Datasets used in evaluations tend to be small-scale and old fashioned. The authors should explore some new challenging datasets to validate the effectiveness of the proposed method, such as [A]. It would be more interesting to see how the proposed model reflect on both scenario-level and anomaly-type-level evaluations.

[A] L Zhu, L Wang, A Raj, T Gedeon, and C Chen. Advancing Video Anomaly Detection: A Concise Review and a New Dataset. Advances in Neural Information Processing Systems (NeurIPS). 2024.

- The evaluations tend to be limited. (i) Lack of enough hyperparameters evaluations and exploration on different model variants to show the effectiveness and robustness. (ii) Most experimental results are presented in the form of tables, it is suggested to use more visualisations such as plots, figures, attention visualisations to show the effectiveness of the proposed method compared to existing methods in the literature.

-Fig 3 presented some visualisations, but still, very limited. On the other hand, it is suggested to also plot eg how existing state of the art methods perform on the qualitative results (plots, proposed method vs ground truth vs existing SOTAs). The discussions and analysis also tend to be a bit limited and it is suggested to improve.

**Questions:**

Refer to weaknesses, and also some questions below.

- Pay more attention to punctuations, eg, quotation marks should appear in pairs in the whole paper (eg, Line 16-17).

- Algorithm 1 is also unclear, as the steps are not clearly presented (one step per sentence).

- Limitations and future works could be provided.

---

### Note · Authors · 2025-12-03

I have read and agree with the venue's withdrawal policy on behalf of myself and my co-authors.